# Targeting Glucose Transporters for Breast Cancer Therapy: The Effect of Natural and Synthetic Compounds

**DOI:** 10.3390/cancers12010154

**Published:** 2020-01-08

**Authors:** Ana M. Barbosa, Fátima Martel

**Affiliations:** 1Instituto de Ciências Biomédicas Abel Salazar, University of Porto, 4169-007 Porto, Portugal; anamargabarbosa@gmail.com; 2Unit of Biochemistry, Department of Biomedicine, Faculty of Medicine, University of Porto, 4200-319 Porto, Portugal; 3Instituto de Investigação e Inovação em Saúde, University of Porto, 4200-135 Porto, Portugal

**Keywords:** breast cancer, glucose transport, drugs, natural compounds

## Abstract

Reprogramming of cellular energy metabolism is widely accepted to be a cancer hallmark. The deviant energetic metabolism of cancer cells-known as the Warburg effect-consists in much higher rates of glucose uptake and glycolytic oxidation coupled with the production of lactic acid, even in the presence of oxygen. Consequently, cancer cells have higher glucose needs and thus display a higher sensitivity to glucose deprivation-induced death than normal cells. So, inhibitors of glucose uptake are potential therapeutic targets in cancer. Breast cancer is the most commonly diagnosed cancer and a leading cause of cancer death in women worldwide. Overexpression of facilitative glucose transporters (GLUT), mainly GLUT1, in breast cancer cells is firmly established, and the consequences of GLUT inhibition and/or knockout are under investigation. Herein we review the compounds, both of natural and synthetic origin, found to interfere with uptake of glucose by breast cancer cells, and the consequences of interference with that mechanism on breast cancer cell biology. We will also present data where the interaction with GLUT is exploited in order to increase the efficiency or selectivity of anticancer agents, in breast cancer cells.

## 1. Introduction

According to the last Global Cancer Statistics (GLOBOCAN 2018), breast cancer represented 12% of all cancers, being the second most frequent cancer worldwide, after lung cancer, and caused about 7% of the total cancer deaths in 2018 [1]. In women, breast cancer is the leading type of cancer and the leading cause of cancer death worldwide [1].

Screening programs and adjuvant chemotherapy have had a significant impact on the prognosis of breast cancer patients, having significantly improved their overall survival, disease-free survival, and death rates related to breast-cancer since the early 1990s [2,3]. Nevertheless, efforts must continue in order to reduce not only the incidence but also the mortality and treatment-associated morbidities associated with this disease. In this context, discovery of new molecular targets and the refinement of lead compounds constitute a priority in breast cancer research.

## 2. Metabolic Reprogramming in Cancer Cells

Metabolic reprogramming and altered energetics is firmly established as a hallmark of cancer and constitutes an active area of basic, translational, and clinical cancer research in recent years [4].

One of the cancer metabolic hallmarks is a deviant energetic metabolism-known as the Warburg effect-characterized by a very high rate of glycolysis and production of lactate, even in the presence of oxygen [5]. Cancer cells have a high dependence on the glycolytic pathway to supply their need of high amounts of adenosine triphosphate (ATP) and also of metabolic intermediates that contribute to several biosynthetic pathways, crucial for cancer progression [4] and to compensate for excess metabolic production of reactive oxygen species (ROS) [6]. So, they shift their main ATP-producing process from oxidative phosphorylation to glucose fermentation, even in aerobic conditions [4]. This altered metabolism may be not only a consequence of genetic mutations, but also a contributing factor or cause of tumorigenesis [7].

More recently, a ‘two-compartment’ model, also named ‘the reverse Warburg effect’ or “metabolic coupling”, has been proposed to reconsider metabolism in tumors, because it was realized that another type of metabolism occurs in certain types of cancers, having high mitochondrial respiration and low glycolysis rate [8]. According to this model, tumor cells and adjacent stromal fibroblasts form a two-compartment model of cancer metabolism, in which fibroblasts perform aerobic glycolysis (because of the acidic microenvironment induced by cancer cells), and the generated metabolites (such as pyruvate, ketone bodies, fatty acids, and lactate), are transferred to tumor cells, to fuel the Krebs cycle and maintain ATP generation [9]. This metabolic coupling is found in some forms of breast cancer [10], and may contribute drug resistance and therapeutic failure in some types of cancers [11], as observed with tamoxifen-resistance in breast cancer MCF7 cells [12].

## 3. Upregulation of Glucose Transport in Breast Cancer Cells

Since the energetic metabolic shift in cancer cells produces less ATP per glucose molecule, the demand for glucose in these cells is higher than in normal cells. Therefore, cancer cells rely on higher rates of glucose uptake in order to support their increased energy, biosynthesis and redox needs. This increased rates of cellular uptake of glucose is met by overexpression of glucose transporters, which is observed in most cancer cells [13].

Two families of glucose transporters mediate glucose uptake in mammalian cells: the Na^+^-dependent glucose co-transporters (SGLTs) and the facilitative glucose transporters (GLUTs).

The SGLT family (gene symbol *SLC5A*) are secondary active transporters that transfer glucose against its concentration gradient coupled with Na^+^ transport down its concentration gradient, which is maintained by the Na^+^/K^+^ pump. For every glucose molecule that is transported, two Na^+^ are also transported. SGLT transporters have 14 transmembrane domains and a high affinity for glucose. At physiological extracellular Na^+^ concentration and membrane potential, an apparent Km of 0.5 mM of SGLT1 for glucose was described, but glucose is transported with a lower affinity when the plasma membrane is depolarized and/or the extracellular Na^+^ concentration is low [14,15]. SGLT1 and SGLT2 overexpression is present in some types of cancer, such as pancreas, prostate, lung, liver, and ovarian cancer, but these transporters have not been described in breast cancer [16].

The GLUT family (gene symbol *SLC2A*) are facilitative transporters that mediate the transport of glucose down its concentration gradient. This family of transporters is composed of 14 members: GLUT1-GLUT12, GLUT14, and the H^+^/myo-inositol transporter. All GLUTs are predicted to have 12 transmembrane domains connected by hydrophilic loops. Each of the GLUT transport protein possesses different affinities for glucose and other hexoses such as fructose. GLUT1, GLUT3, and GLUT4 have a high affinity for glucose (e.g., the Km of GLUT1 for glucose is 1–3 mM), allowing transport of glucose at a high rate under normal physiological conditions [17].

Increased cellular uptake of glucose in tumor cells is associated with increased and deregulated expression of GLUT transporters [13]. Among GLUT family members, overexpression of GLUT1 has been consistently observed in many different cancers, including breast, lung, renal, colorectal, and pancreatic cancers [13,18,19]. Consistent with its overexpression, GLUT1 is crucial for uptake of glucose by breast cancer cells [20,21,22] and is also the main glucose transporter in breast cancer cell lines (e.g., MCF-7 and MDA-MB-231) [21,23]. GLUT1 is a transporter ubiquitously expressed in most mammalian tissues (abundantly in brain and erythrocytes), being responsible for basal glucose cellular uptake in the majority of tissues [16,17]; it is also the predominant isoform present in human and bovine mammary glands [24,25]. Glucose uptake mediated by GLUT1 appears to be especially critical in the early stages of breast cancer development, affecting cell transformation and tumor formation [26,27]. Indeed, GLUT1 overexpression, which occurs early during the transformation process, induces a change in breast epithelial cell metabolism that precedes morphological changes in breast cancer, and thus may be a fundamental part of the neoplastic process [18]. Interestingly, the loss of even a single GLUT1 allele is sufficient to impose a strong break in breast tumor development in a mouse model [26]. A strong correlation between *GLUT1* gene expression and breast cancers of higher grade and proliferative index and lower degree of differentiation [28] and higher malignant potential, invasiveness, and consequently poorer prognosis [29] exists. GLUT1 is thus considered an oncogene [18,19,20,30].

One of the factors responsible for the upregulation of GLUT1 in breast tumor cells is hypoxia. The promoters of GLUT1 contain hypoxia-response elements, which bind the hypoxia-inducible factor (HIF-1) to facilitate transcription. Since an increase in the levels of HIF-1α protein is a phenomenon seen in most cancers, it provides a molecular mechanism for cancer-associated overexpression of GLUT1 [18,31]. Additionally, hypoxia appears to increase GLUT1 transport activity in the MCF-7 breast cancer cell line, independently of changes in transporter expression [32]. Besides HIF-1, the ovarian hormone estrogen is also known to induce GLUT1 expression in breast cancer [18,33]. Moreover, the histone deacetylase SIRT6, the cellular oncogene product c-MYC (V-Myc Avian Myelocytomatosis Viral Oncogene Homolog), the pro-survival protein kinase Akt (Protein Kinase B) and mutant p53, all of which induce the expression of GLUT1 [31,34], can also be involved in GLUT1 overexpression in breast cancer.

In addition to GLUT1, which is consistently found to be expressed in breast tumors and cell lines, other GLUT family members can also contribute to glucose uptake by breast cancer cells. More specifically, GLUT2 [19,23] and GLUT3 [18] are also expressed in several breast cancer cell lines. Additionally, GLUT4 expression [30,35,36,37] and insulin-stimulated glucose uptake were also described in some cancer cell lines [38,39,40]. Moreover, the involvement of GLUT4 in basal glucose uptake was described in two breast cancer cell lines [41]. Finally, a second insulin-stimulated transporter, GLUT12, was also described in MCF-7 cells [18,42]. Similar to GLUT1, the expression of GLUT3 and GLUT12 correlate with poor prognosis [18,19]. Importantly, increased expression of GLUT1 and GLUT3 was also associated with resistance of cancer cells to radio or chemotherapy [43,44,45], but the underlying mechanisms linking GLUT and chemo- or radio-resistance remain largely unknown.

Increased glucose uptake by cancer cells has been exploited clinically in diagnosis and follows up of cancer via the use of ^18^fluoro-2-deoxy-D-glucose (FDG), a radiolabeled glucose analogue, in Positron Emission Tomography (PET) [46]. This radiotracer enters cells via GLUTs, being then phosphorylated by hexokinases into FDG-6-phosphate that cannot be further metabolized and thus accumulates in the cytoplasm. Importantly, the sensitivity of this technique varies depending on the type of cancer, and this heterogeneity has been particularly associated with GLUT1 or GLUT3 tumor expression [23,47].

## 4. Glucose Transporters as Therapeutic Targets in Breast Cancer

Since cancer cells depend on increased utilization of glucose as compared to normal healthy cells, glucose deprivation is considered an effective anticancer therapy and as a potential strategy for cancer prevention, and many compounds targeting cancer cell energy metabolism are currently on trial or approved as therapeutic agents against cancer [48,49]. These include specific inhibitors of monocarboxylate transporter 1, hexokinase II, glyceraldehyde-3-phosphate dehydrogenase (GAPDH), pyruvate dehydrogenase, pyruvate dehydrogenase kinase 1, cancer-specific mutant isocitrate dehydrogenase, lactate dehydrogenase A, phosphoglycerate mutase 1, phosphofructokinase, or pyruvate kinase M2 [48,50]. In support of glucose deprivation as a molecular target in cancer, high-fat and low-carbohydrate diet appear to provide therapeutic benefits for increased survival by reducing angiogenesis, peri-tumoral edema, cancer migration, and invasion [51]. According to some authors, inhibition of glucose metabolism will not only deplete cancer cells of ATP, but also will lead to enhanced oxidative stress-related cytotoxicity [6].

Additionally, because tumor cells have an increased dependence in relation to extracellular glucose, GLUTs constitute also an anticancer target [18,52,53,54]. A direct approach to this therapeutic target is to block GLUT-mediated glucose uptake, which would abolish entry of glucose into the cancer cell. Alternatively, new approaches consist in the design and development of “GLUT-transportable anticancer agents”, or the use of GLUT antibodies to selectively deliver an anticancer agent to cancer cells.

In this review, we will list compounds, both of natural and synthetic origin, found to interfere with glucose uptake by breast cancer cells, and present the consequences of GLUT inhibition and/or knockout on breast cancer cell biology. We will also present data where the interaction of defined molecules with GLUT is exploited in order to increase its efficiency or selectivity, in breast cancer cells.

## 5. Effect of Synthetic and Natural Compounds on Glucose Uptake by Breast Cancer Cells

### 5.1. Effect of Synthetic Compounds

#### 5.1.1. GLUT Inhibitors

##### WZB117 and STF-31

The effect of two recently described GLUT1 inhibitors, WZB117 and STF-31, on breast cancer cells was studied by some authors. WZB117 is a representative of a group of novel small compounds that were recently reported to inhibit basal glucose transport and cell growth in vitro and in vivo [55,56]. STF-31 is a small molecule that was firstly reported to selectively target von Hippel-Lindau (VHL)-deficient renal cell carcinoma (RCC) cells [55].

These two GLUT1 inhibitors were able to inhibit cell proliferation and induce apoptosis in several breast cancer cell lines (MCF-7, MDA-MB-231, HBL100, and BT549), and these effects were accompanied by interference with cellular glucose handling, increasing the levels of extracellular glucose, and decreasing the levels of extracellular lactate, suggesting an inhibitory effect upon glucose uptake and/or glycolysis. Of interest, STF31 (30 µM) potentiated the antiproliferative effect of metformin (3 mM) in MDA-MB-231 cells [57]. Although the effect on glucose uptake was not studied, GLUT1 inhibition (with WZB117) blocked transformation of MCF10A-ERBB2 cells (a breast epithelial cell line used as a model to study the early events leading to transformation) induced by activated ERBB2 through reduced cell proliferation [26] (Table 1).

In addition of testing these GLUT1 inhibitors alone as a targeted therapy, GLUT1 inhibition in combination with other cancer therapeutics has also been evaluated (Table 1). In one study, WZB117 was found to reduce GLUT1 mRNA and protein levels and glucose uptake and lactate production in two breast cancer cell lines (MCF-7 and MDA-MB-231). The interaction of this agent with radiation was investigated. Glucose metabolism and GLUT1 expression were found to be significantly stimulated by radiotherapy. Interestingly, radioresistant breast cancer cells exhibited upregulated GLUT1 expression and glucose metabolism but combination of WZB117 and radiation re-sensitized the radioresistant cancer cells to radiation [58]. A synergic antitumoral effect was also found between WZB117 and the anticancer drugs cisplatin and paclitaxel, in MCF-7 cells [59]. Finally, the possibility that a combined treatment with a GLUT1 inhibitor could overcome resistance to another breast cancer therapeutic agent (adriamycin) was also investigated. Resistance to adriamycin is a common obstacle occurring during therapy of breast cancer patients. WZB117 was found to resensitize MCF-7/ADR cells (adriamycin-resistant) to adriamycin [60]. Therefore, GLUT1 inhibition could overcome resistance to adriamycin and radiation.

##### WZB27 and WZB115

Two other GLUT1 inhibitors, WZB27 and WZB115, were synthesized and tested against several cell types, including a breast cancer cell line (MCF-7). These compounds reduced basal glucose uptake and cell proliferation, induced apoptosis, and led to cell cycle arrest in G1/S phase, without affecting much the normal cell line MCF12A. Importantly, their inhibitory effect on cancer cell growth was ameliorated when additional glucose was present, suggesting that the inhibition was due, at least in part, to inhibition of basal glucose uptake. Moreover, when used in combination, the test compounds demonstrated synergistic effects with the anticancer drugs cisplatin and paclitaxel (Table 1) [56].

##### Bay876

BAY-876 is a highly selective GLUT1 inhibitor under preclinical study for oncolytic treatment [61]. In a recent report, the interaction between GLUT1 and bromodomains (BRDs) was investigated. BRDs are conserved protein interaction modules, which recognize acetyl-lysine modifications, and BRD-containing proteins are components of the transcription factor and chromatin-modifying complexes and determinants of epigenetic memory [62]. BAY876 decreased glucose uptake by a triple-negative breast cancer cell line, and a vulnerability of these breast cancer cells to inhibition of BRPF2/3 BRDs, under conditions of glucose deprivation or GLUT1 inhibition, was reported (Table 1) [63].

##### 2-deoxy-D-glucose

2-deoxy-D-glucose (2-DG) is a synthetic non-metabolizable glucose analogue. 2-DG inhibits the glycolytic pathway, because the product of its phosphorylation by hexokinase cannot be further metabolized and, additionally, is a non-competitive inhibitor of hexokinase, thus causing ATP depletion. Additionally, 2-DG competes with glucose for GLUT [64]. In a triple-negative breast cancer cell line (MDA-MB-231), but not in an estrogen receptor (ER)-positive cell line (MCF-7), 2-DG was able to reduce glucose uptake (Table 1) [65].

##### GLUT1 shRNA

Another strategy that is being tested to target GLUT1 is by RNA interference (RNAi) using short hairpin RNA (shRNA). Silencing of GLUT1 expression with an shRNA led to a significant decrease in glucose uptake in vitro in both a triple-negative (MDA-MB-468) and a HER2-positive cell line (SK-BR3), together with a decrease of the growth of xenograft tumors (MDA-MB-468 cells) [66]. Similarly, shRNA targeting GLUT1 decreased glucose transport and consumption, reduced lactate secretion, and inhibited growth of the mouse mammary tumor cell line 78617GL, both in vitro and in vivo (Table 1) [27].

A similar negative effect of GLUT1 shRNA on glucose uptake was found in two other triple-negative breast cancer cell lines (MDA-MB-231 and Hs578T), together with a decrease in cell proliferation, migration, and invasion, which was concluded to result from GLUT1-mediated modulation of Epidermal Growth Factor Receptor (EGFR)/Mitogen-Activated Protein Kinase (MAPK), and integrin β1/Src/FAK signaling pathways [67]. However, the same group verified that, contrary to the expected, ablation of GLUT1 attenuated apoptosis and increased drug resistance in triple-negative breast cancer cells (MDA-MB-231 cells), via upregulation of p-Akt/p-GSK-3β (Ser9)/β-catenin/surviving (Table 1) [52]. Not only is the prognostic of triple-negative breast cancer (TNBC) usually poor due to aggressive tumor phenotypes, but also because conventional chemotherapy cannot be used. Therefore, and because TNBC have higher levels of GLUT1, this transporter is seen as a potential therapeutic target, sensitizing cells to chemotherapy. The results of this later study, however, indicate that the potential of GLUT1 as a therapeutic target in TNBC should be carefully re-evaluated [68].

##### Anti-GLUT1-antibody

An anti-GLUT1 monoclonal antibody was able to decrease glucose uptake in breast cancer cells (MDA-MB-231), and to reduce cell proliferation and stimulate apoptosis (MCF-1 and T47D). Importantly, when associated with chemotherapeutic agents (5 μM cisplatin, 5 μM paclitaxel, or 10 μM gefitinib), it potentiated the anti-proliferative and pro-apoptotic effects of these agents in MCF-7 cells (Table 1). The authors concluded that the use of antibodies to GLUT1 may be a viable but an as yet unexplored therapeutic strategy in tumors that overexpress GLUT1 [69].

##### GLUT4 shRNA

By stably silencing GLUT4 expression by lentiviral expression of a GLUT4 shRNA, GLUT4 was concluded to have a prominent role in basal glucose uptake in MCF7 and MDA-MB-231 breast cancer cells (Table 1). Moreover, GLUT4 specific downregulation in these two different breast cancer cell lines, with different degrees of malignancy and differentiation, promoted metabolic reprogramming and affected cell proliferation and viability. According to these authors, their study provides proof-of-principle for the feasibility of using pharmacological approaches to inhibit GLUT4 in order to induce metabolic reprogramming in vivo in breast cancer models [41].

#### 5.1.2. Antidiabetics

Biguanides, including metformin and phenformin, are inhibitors of mitochondrial respiratory chain complex I, and have been shown to reduce cancer incidence and cancer-related death [68].

##### Metformin

Metformin is the most prescribed oral antidiabetic drug used for the treatment of diabetes mellitus. Metformin was associated with reduced risk of developing cancer in diabetic patients in 2005 [70]. Since then, a large amount of studies confirmed this observation, and the role of metformin in breast cancer has been evaluated [71,72]. In this context, the effect of metformin on glucose uptake and metabolism by breast cancer cells has been evaluated in a few studies (Table 1).

In a first study, metformin was found to decrease glucose utilization both in vitro (MDA-MB-231) and in vivo (using MDA-MB-231 cells orthotopically implanted in a mammary fat pad). However, this effect was concluded to be related to a direct inhibitory effect on the glycolytic enzyme hexokinase and drug effects on transmembrane glucose transport were excluded, because glucose uptake and glucose transporters expression levels were not affected by metformin [73].

In the study by Amaral et al. [65], short-term exposure to metformin inhibited glucose uptake, probably by direct inhibition of GLUT1, and, in contrast, long-term exposure to metformin led to a significant increase in glucose uptake, which was not associated with changes in GLUT1 mRNA levels. It was suggested that the increase in glucose uptake induced by long-term metformin, is a compensatory mechanism in response to cellular ATP depletion resulting from its inhibitory effect on oxidative phosphorylation and that this metformin-induced dependence on glycolytic pathway, associated with an anticarcinogenic effect of the drug, provides a biochemical basis for the design of new therapeutic strategies. The increase in glucose uptake after a long-term exposure to metformin, to compensate for the reduced mitochondrial ATP generation, was corroborated in another study, using two triple-negative breast cancer cell lines (MDA-MB-231 and MDA-MB-436) [74]. Lastly, the interaction between metformin and PPARδ (peroxisome-proliferator-activated receptor δ), known to have a role in inflammation, metabolism, and cancer, was recently evaluated [75]. Metformin was able to block the increase in GLUT1 and SGLT1 mRNA and protein levels, glucose uptake, glucose consumption, and lactate production caused by the PPARδ agonist GW501516 in MCF-7 cells. The effect of metformin in reducing the expression of GLUT1 and SGLT1 was not present with metformin alone; rather, it results from metformin-mediated inhibition of PPARδ activity. Therefore, metformin can block the effect of GW501516, but has no effect of its own in reduction of glucose transporters levels, which is in concordance with the previous studies.

##### Phenformin

A recent study showed that glucose uptake and utilization affects cancer cell sensitivity to phenformin treatment. More specifically, a correlation between low expression of glucose transporters, including GLUT1, and both a defective glucose uptake/utilization and an increased sensitivity to phenformin treatment was found in several cancer cell lines. Moreover, restoration of GLUT1 expression attenuated the phenformin-sensitivity in the corresponding cancer cells [76]. Additionally, Liu and Gan [77], by using the MDA-MB-231 cell line, demonstrated that phenformin upregulates GLUT1 levels, causing increased glucose uptake and production of lactate. Importantly, they verified that this effect of phenformin is dependent on NBR2 (neighbor of *BRCA1* gene 2), a glucose starvation-induced long non-coding RNA that interacts with AMP-Activated Protein Kinase (AMPK) and regulates AMPK activity. They thus concluded that the NBR2-GLUT1 axis may serve as an adaptive response in breast cancer cells to survive in response to phenformin treatment (Table 1) [77].

##### Troglitazone

Another antidiabetic drug also associated with an anticarcinogenic effect [78], troglitazone, belongs to the class of thiazolidinediones, which activate peroxisome proliferator-activated receptor-γ (PPARγ), although it has been withdrawn from the market due to its hepatotoxicity. Unlike the antidiabetic effects of this drug, many other actions of troglitazone are thought to occur in a PPARγ-independent manner. Since troglitazone is known to cause mitochondrial dysfunction, its effect on glucose metabolism was investigated [79]. Troglitazone enhanced uptake of glucose in several breast cancer cell lines, but changes in GLUT levels do not seem to play a role in this effect, that rather appears to involve MAPK, AMPK, and EGFR. Interestingly, troglitazone reduced T-47D cell content, and this effect was potentiated by restriction of glucose availability. So, it was concluded that troglitazone stimulates uptake of glucose by cancer cells and shifts its metabolism toward glycolysis, likely as an adaptive response to impaired mitochondrial oxidative respiration (Table 1) [80].

#### 5.1.3. Chemotherapeutic Agents

##### Cisplatin

Cisplatin (cis-diamminedichloroplatinum II) is a very common used chemotherapeutic agent. It is a platinum-derived agent that interferes with DNA replication, and has also been associated with mitochondrial damage. Wang et al [36] used the MDA-MB-231 cell line in order to study cisplatin’s metabolic effects. The compound decreased glucose uptake and lactate production and the expression levels of GLUT1 and GLUT4 (Table 1). Cisplatin downregulation of integrin β5 (ITGB5)/FAK signaling pathway was concluded to be responsible for its effect on the expression of GLUT1 and GLUT4 [36].

###### Sorafenib

The bisarylurea sorafenib is a multi-kinase inhibitor with anti-proliferative and anti-angiogenic activity, currently under evaluation in a variety of solid tumors. Evidence has shown that sorafenib can inhibit oxidative phosphorylation in some types of cancer cell lines [80,81], and the question if it also affects glucose metabolism was then addressed [82]. In this work, the effect of sorafenib on glucose uptake, utilization, lactate production, and GLUT1 expression was investigated in several breast cancer cell lines. Sorafenib produced distinct early and long-term effects on glucose uptake, metabolism, and GLUT1 expression in MCF-7 (ERα-positive), MDA-MB-231 (triple negative), and SKBR3 (ERα-negative/HER2-positive) cell lines. Fasentin (a GLUT1 inhibitor) inhibited the initial GLUT1 overexpression caused by sorafenib and, importantly, its cytotoxic effect (Table 1). It was concluded that the early-term effects were dependent on AMPK and thought to compensate for the loss of mitochondrial ATP, but that persistent activation of AMPK by sorafenib finally led to the impairment of glucose metabolism in all the cell lines, resulting in cell death [82].

###### Trastuzumab

Trastuzumab, effective in about 15% of women with breast cancer, targets Human Epidermal Growth Factor Receptor 2 (HER2) and downregulates signaling through Akt/phosphoinositide 3-kinase (PI3K) and MAPK pathways. These pathways modulate glucose metabolism and so it was evaluated if trastuzumab decreased glucose uptake in breast cancer cells. For this, xenografts derived from HER2-overexpressing MDA-MB-453 human breast tumor cells were grown in severe combined immunodeficient mice. Xenografts were significantly smaller and [^18^F] FDG uptake was also reduced in trastuzumab-treated mice. This observation was accompanied by lower GLUT1 protein levels (Table 1) [83].

###### Doxorubicin (DOX) and 5-fluorouracil (5FU)

The effect of these two chemotherapeutic agents on the expression and activity of GLUT1 and hexokinase and on glucose uptake by the MCF-7 breast cancer cell line was evaluated [84]. Both agents induced a decrease in glucose uptake together with an increase in GLUT1 mRNA levels. The effect on GLUT1 protein levels were not as marked, which suggest posttranslational alterations in GLUT1. It was concluded that after DOX or 5FU therapy, the relationship between glucose and viable cell number can become disjointed, with transient declines in glucose uptake in excess of the decline in cell number despite increased GLUT1 mRNA levels [84]. In another work, DOX and selenium, either free or in PLGA (poly (d, l-lactide-co-glycolide)) nanoparticles were described to reduce the cellular uptake of glucose by MCF-7 and MDA-MB-231 cells, based on measurements of medium glucose levels [85]. So, no direct measurement of glucose uptake was made (Table 1).

###### Palbociclib

Dysregulation of the cell cycle is a hallmark of cancer that leads to aberrant cellular proliferation and inhibition of cell cycle regulators such as Cyclin-Dependent Kinase 4 (CDK4) and 6 (CDK6) has become a new therapeutic target for the treatment of breast cancer. Palbociclib, an orally-available inhibitor of CDK4 and CDK6, represents the most widely studied compound among cell cycle inhibitors [86]. Interestingly, palbociclib also seems to be able to inhibit GLUT1 mediated glucose uptake and metabolism in TNBC cells [87,88]. Moreover, combination of palbociclib with a chemotherapeutic agent currently used for the treatment of TNBC patients (paclitaxel) inhibited cell proliferation and increased cell death more efficiently than single treatments, associated with a more marked effect on glucose uptake and consumption and on GLUT1 protein levels [87]. Additionally, combination of palbociclib with a PI3K/mTOR inhibitor (BYL719) enhanced the antitumoral effect of these agents and the negative effect of each of these drugs on glucose uptake and consumption and on GLUT1 protein levels (Table 1) [88].

#### 5.1.4. Other Drugs

##### Propranolol

Another type of drug that has been getting attention for its recently found anticarcinogenic effect are beta-blockers, more specifically propranolol (PROP). Clinical evidence has strongly indicated that PROP can inhibit cancer growth, metastasis development, and tumor recurrence in breast cancer patients [89]. Treatment with PROP decreased hexokinase-2 expression in vitro and 18F-FDG uptake in vivo, but GLUT1 levels were not affected. This indicates that GLUT1 is not involved in the anticarcinogenic effect of PROP (Table 1) [90].

##### Saracatinib

Prevention of estrogen receptor negative (ER-) and tamoxifen-resistant (TamR) breast cancer remains an important demand due to gaps in pathobiological understanding of this type of cancer. Transforming (sarcoma-inducing) Gene of Rous Sarcoma Virus (Src) activation appears to be a key signaling event driving ER- and TamR breast cancer progression and thus, targeting Src may prevent ER- breast cancer [91]. Accordingly, Src-targeting agents such as the tyrosine kinase inhibitor saracatinib, have been extensively tested in the clinic for treatment of metastatic breast cancer [92]. In the report by Jain et al. [93], activation of Src kinase was investigated as an early signaling alteration in premalignant breast lesions of women who did not respond to tamoxifen, a widely used ER antagonist for hormonal therapy of breast cancer. They verified that Src plays an essential role in regulating glucose uptake, because knocking down Src significantly reduced glucose uptake. Moreover, they showed that saracatinib inhibited glucose uptake in premalignant breast cell lines (MCF-10A and MCF12A) with or without HER-2 overexpression (Table 1).

##### P53 Modulators

The tumor protein p53, a well-recognized tumor suppressor, is a key regulator of energy metabolism, playing an important role in preventing the cell from reprogramming its energetic metabolic pathway [94]. The p53-reactivating compound RITA (Reactivating p53 and Inducing Tumor Apoptosis) activates p53 in cells expressing oncogenes, whereas its effect in non-transformed cells is almost negligible [95]. This agent decreased GLUT1 mRNA expression in MCF-7 cells. Further, another p53 activator, nutlin3a [96], also caused repression of GLUT1 mRNA expression. Interestingly, the p53 inhibitor pifithrin-α p53 [95] induced the expression of GLUT1 mRNA and abolished the effect of RITA upon GLUT1 mRNA levels (Table 1) [94]. This study shows that reinstatement of p53 function targets the dependence of cancer cells on glycolysis, which can contribute to the selective killing of cancer cells by pharmacologically activated p53.

##### Akt Inhibitors

The protein kinase Akt is involved in various cellular processes, including cell proliferation, growth and metabolism, and hyperactivation of Akt is commonly observed in human tumors [97]. Three non-ATP-competitive allosteric Akt inhibitors (Akt1i, Akt2i, and Akt1/2i) reduced glucose transport into T-47D breast cancer cells, by interfering with a process distinct from the Akt signaling pathway (involved in movement of GLUT4 to the plasma membrane, e.g., in adipocytes). Among other evidences, the PI3K inhibitor wortmannin was devoid of effect on glucose uptake. It was concluded that these drugs, at least in part, inhibit tumorigenesis through inhibition of glucose transport in tumor cells (Table 1) [98].

##### PGC1β and HKDC1 shRNA

The peroxisome proliferator-activated receptor-γ (PPARγ) co-activator-1b (PGC1b) promotes tumorigenesis by modulation of mitochondrial function and glycolysis metabolism [99]. On the other hand, hexokinase domain component 1 (HKDC1), recently discovered as a putative hexokinase [100], may be a novel potential therapeutic target for cancer [101]. A recent study demonstrated that knockdown of either PGC1β or HKDC1 resulted in a decrease in glucose uptake in MCF-7 cells (Table 1) [102].

##### miRNA-34a Inhibitor

miRNA-34a is a tumor suppressor that is expressed in a variety of different types of cancer, including breast cancer. A recent report showed that miRNA-34a inhibition promoted cancer cell proliferation, accelerated glucose uptake and upregulated GLUT1 expression in two triple-negative breast cancer cell lines used, but interestingly, was devoid of effect in the normal human breast epithelial cell line (Table 1) [103].

##### miRNA-186-3p

Recently, it was verified that systemic delivery of cholesterol-modified agomiR-186-3p to mice bearing tamoxifen-resistant breast tumors effectively attenuates both tumor growth and ^18^F-FDG uptake (Table 1) [104].

### 5.2. Effect of Endogenous Compounds

#### 5.2.1. Hormones

##### Melatonin

Melatonin is produced and secreted by the pineal gland, and modulates several biological pathways in cancer [105]. Interestingly, GLUT1 appears to be involved in the uptake of melatonin into cancer cells and melatonin appears to bind the glucose binding site of the transporter [106]. Despite the lack of reports elucidating the effects of melatonin targeting the Warburg effect in breast cancer cells, an important study using a xenograft breast cancer model found that glucose uptake and lactate production were inversely correlated with melatonin levels during the 12:12 light:dark cycle [107]. In this context, a recent report evaluated the effect of low pH (6.7) on human breast cancer cell lines (MCF-7 and MDA-MB-231), and the effectiveness of melatonin in the acid tumor microenvironment. Melatonin was able to decrease GLUT1 protein expression levels in both cell lines, both at normal (7.2) and acidic pH (6.7). It was concluded that melatonin treatment increases apoptosis and decreases proliferation and GLUT1 protein expression under acute acidosis conditions in breast cancer cell lines [108] (Table 2).

##### 17β-oestradiol

17β-oestradiol or E2 is a steroid hormone, being the main female sexual hormone. Besides its physiological effects, such as maintenance of reproductive cycle and secondary female characteristics, it plays a major role in the carcinogenesis of breast cancer. A few studies evaluated the effect of E2 on glucose uptake be breast cancer cells (Table 2). In a study using MCF-7 cells, E2 was concluded to have no effect on glucose cellular uptake. In this study, E2 increased culture growth, proliferation rates, cellular viability, and lactate production, but did not affect the uptake of glucose nor GLUT1 mRNA levels. So, it was concluded that the pro-proliferative and cytoprotective effects of E2 are not dependent of stimulation of glucose cellular uptake [109]. In contrast, previous studies found E2 to increase the rate of glucose utilization (although glucose uptake levels were not really measured) [110], to increase glucose uptake and expression/translocation of GLUT4 into the plasma membrane (although E2 showed no effect on the expression/translocation of GLUT1) [30,35], and to increase the expression of GLUT1 [33], or to have no effect on GLUT1 expression levels, although higher rates of glucose uptake were found [111]. So, the effect of E2 on glucose uptake and transporter expression needs to be further clarified. Of note, in the work of Rivenzon-Segal et al. [33], tamoxifen had an opposite effect on GLUT1 and treatment of the cells with both E2 and tamoxifen resulted in a partial (±50%) abolishment of the effect of E2 on GLUT1, demonstrating thus the antiestrogenic activity of tamoxifen with regard to GLUT1 expression.

##### Progesterone

In the work by Medina et al. [30] the effect of progesterone, which also increases the risk of breast cancer, on GLUT expression levels and glucose uptake by ZR-75-1 cells was analyzed (Table 2). This hormone was found to increase glucose uptake and the expression levels of GLUT1, GLUT3 and GLUT4 [30].

##### Glucocorticoids

Stress has a vast variety of effects in the human body, one of them being the stimulation of the production of adrenocorticotropic hormone by the anterior pituitary gland. This increases secretion of glucocorticoids, steroid hormones that modulate inflammation and the immune system, cell differentiation, and metabolism [112]. Glucocorticoids are often prescribed in chemotherapy treatments in order to avoid hypersensitivity reactions. Therefore, it is important to investigate if it affects treatment. Additionally, glucocorticoids such as dexamethasone may be involved in resistance processes (chemotherapy desensitization) in various types of solid neoplasms, including breast cancer [113]. Dexamethasone (10.7–10.8 µM; 3 days) showed antiproliferative properties on MCF-7 cells [114]. The antiproliferative effect of dexamethasone in MCF-7 cells was confirmed in a later study; this effect was associated with a slight increase in glucose uptake, a strong increase in GLUT4 expression levels and with the formation of adipocyte-like vesicles. In contrast, dexamethasone did not affect MDA-MB-231 cells proliferation, although it slightly increased glucose uptake and strongly increased GLUT4 expression levels. The authors concluded that dexamethasone treatment induces inhibition of cell growth of dexamethasone-sensitive cancer cells by stimulation of differentiation into adipocyte-like cells [112] (Table 2).

##### KL1

Klotho is a transmembrane protein that can be shed and act as a circulating hormone in three forms: soluble klotho, KL1, and KL2 [115]. Klotho was proposed to be implicated in aging through inhibition of the Insulin-like Growth Factor 1 (IGF-1) pathway, but it also functions as a tumor suppressor in several types of cancer, including breast cancer [116]. This hormone was recently found to decrease glucose uptake and glycolytic flux in MCF-7 cells, but the mechanism of action was not reported [117] (Table 2).

##### Insulin

Insulin a peptide hormone secreted by the β cells of the pancreatic islets of Langerhans, with an important role in the maintenance of glucose blood levels. Additionally, insulin exhibits potent anabolic properties and has been implicated in many malignancies, including breast cancer [118]. Insulin is also known to be a modifier of cancer cell metabolism. Indeed, it regulates carbohydrate and lipid metabolism, stimulates DNA synthesis, modulates transcription [118], and stimulates the cellular uptake of various nutrients, including glucose, by facilitated diffusion [119]. Agrawal et al. studied the effect of insulin on the sensitivity of a breast cancer cell line (MCF-7) to 5-fluorouracil (5FU) and cyclophosphamide (CPA) [120]. The chemotherapeutic agents 5FU and CPA are widely used in the clinic and incorporated in the treatment of several cancer, including breast cancer, being associated with increased levels of chemoresistance. Insulin was found to increase the cytotoxic effects of 5FU and CPA in vitro up to two-fold. This effect of insulin was linked to enhancement of apoptosis, activation of apoptotic and autophagic pathways, and to overexpression of GLUT1 and GLUT3 as well as to inhibition of cell proliferation and motility (Table 2). Therefore, it was concluded that insulin sensitization before chemotherapy treatment could overcome chemoresistance [120]. The effect of insulin upon GLUT1 and GLUT3 protein expression levels were hypothesized to be mediated by the PI3K-Akt pathway, but the hypothesis was not tested.

#### 5.2.2. Other Endogenous Compounds

##### Lactic Acid

Besides being a metabolic fuel, lactate is an important signaling molecule in cancer. This compound induces angiogenesis [121], induces HIF1α, which is associated with cancer cell growth and poor prognosis [122], and stimulates folate uptake by breast cancer cells [38]. Lactic acid interferes also with glucose uptake by breast cancer cells, but either a stimulatory [38] or an inhibitory effect [123] were described (Table 2). In the interesting paper of Turkcan et al, single cells from the core of 4T1 and MDA-MB-231 mice large tumors (>8 mm diameter) grafts were found to take up less glucose than those from the periphery. The authors were able to show that this difference was attributed to an inhibitory effect of lactic acid on glucose uptake [123].

##### Interleukin-4

Cytokines and chemokines in the tumor microenvironment promote breast cancer progression and metastasis. The interleukin-4 (IL4)/IL4Rα immune signaling axis is a direct promoter of survival and proliferation in breast cancer cells [124]. Venmar et al. [124] investigated whether IL4R-mediated metabolic reprogramming could support tumor growth. They verified that promotion of tumor cell survival and proliferation by IL4 involved an increase in glucose uptake and lactate production by murine 4T1 breast cancer cells, associated with an increase in GLUT1 expression, both in vivo and in vitro (Table 2). Moreover, that concluded that, in addition to IL4, there may also be a role for the second IL4Rα-binding cytokine, IL13, in promoting GLUT1 expression through IL4Rα. Importantly, this effect of IL4 on glucose uptake and transporter expression in murine breast cancer cells was not observed in the human MDA-MB-231 cells [124].

##### Epidermal Growth Factor

Breast cancer that expresses epidermal growth factor receptors (EGFR) is associated with poor patient prognosis, both in TNBC and in non-TNBC subtypes [125]. In breast cancer patients, EGFR expression is strongly correlated with tumor uptake of the glucose analogue, ^18^F-FDG [126]. An in vitro study with three breast cancer cell lines showed that Epidermal Growth Factor (EGF) stimulated glucose uptake in EGFR-positive T-47D and MDA-MB-468 cells, but not in the weakly EGFR-positive MCF-7 cells. In T-47D cells, the effect was accompanied by upregulated GLUT1 expression and increased lactate production. EGFR stimulation also increased T47D cell proliferation [127] (Table 2).

### 5.3. Effect of Exogenous Natural Compounds

#### 5.3.1. Polyphenols

A large class of GLUT inhibitors is represented by polyphenols, a heterogeneous and large family of natural compounds widely distributed in plants and in the human diet (e.g., in fruits, vegetables and beverages such as tea and wine) [128]. Many of these compounds show an appreciable activity on several distinct membrane transporters, including GLUTs [129,130]. Polyphenols possess anticancer effects in relation to several cancer types, including breast cancer. Several distinct mechanisms are involved in their anticarcinogenic effect in breast cancer: interference with redox balance, pro-apoptotic effect, cell cycle arrest, activation of autophagy, inhibition of angiogenesis, anti-inflammatory effect, anti-estrogenic effect, changes in ER expression, aromatase modulation, interference with HER2 signaling, and effect on microbiota [131,132]. Additionally, some polyphenols interfere with glucose cellular uptake by breast cancer cells [133,134], as next described.

##### Gossypol

This polyphenolic bisnaphthalene aldehyde obtained from the cotton plant markedly increased both glucose consumption and lactate production in MCF-7 cells, but the increase in glucose consumption may not related to an increase in glucose uptake, and rather be the consequence of increased glycolytic rates or increased rates of glycose oxidation not related to glycolysis (e.g., pentose phosphate pathway) [135] (Table 3).

##### Naringenin

This grapefruit flavanone inhibited both basal and insulin-stimulated glucose uptake in two breast cancer cell lines (MCF-7 and T-47D). The reduction in insulin-stimulated glucose uptake was not associated with changes in GLUT4 protein levels but rather with inhibition of insulin-stimulated PIP3/Akt and p44/p42 MAPK activity [39]. The antiproliferative effect of naringenin was mimicked by low glucose conditions and so it was concluded that it was dependent on impairment of glucose uptake [39] (Table 3).

##### Genistein

The flavonoid genistein, found in soybean, reduced glucose uptake in both estrogen receptor-positive MCF-7 and -negative (MDA-MB-231) breast cancer cell lines [136]. The inhibitory effect of genistein upon glucose uptake by MCF-7 cells was later confirmed in two studies. In the first, the effect of genistein, daidzein, and a soy seed extract on two distinct breast cancer cell lines were investigated. In MCF-7 cells, these compounds presented an inhibitory effect on cell proliferation that correlated with a decrease in glucose cellular uptake [137]. In the second, exposure to several polyphenols, including genistein (myricetin, genistein, resveratrol, and kaempferol), was shown to reduce glucose uptake by MCF-7 cells, and genistein inhibited glucose uptake with a 50% Inhibitory Concentration (IC_50_) of 39 µM [138] (Table 3).

##### Kaempferol

In the work of Azevedo et al. [138], kaempferol was found to be the most potent inhibitor of glucose uptake, with an IC_50_ of 4 µM. Kaempferol (30 µM) decreased glucose uptake and the GLUT1 transcription level. Moreover, low extracellular glucose mimicked, and high extracellular glucose conditions prevented, the antiproliferative and cytotoxic properties of kaempferol. So, it was suggested that inhibition of GLUT1-mediated glucose cellular uptake mediates the anticancer effect of kaempferol in MCF-7 cells [138] (Table 3).

##### Resveratrol

An inhibitory effect of resveratrol (IC_50_ = 67 µM), found in fruits such as grapes and berries, upon glucose uptake by breast cancer cells was also described in the work by Azevedo et al. [138]. Moreover, an inhibitory effect of this stilbene was previously described in another breast cancer cell line. Resveratrol (150 µM), suppressed uptake of glucose and glycolysis in T-47D breast cancer cells, associated with a reduction in GLUT1 expression and dependent on a reduction in intracellular ROS levels, which decreases HIF-1α accumulation [139] (Table 3).

##### Hesperitin

This flavanone, found in citrus fruits, reduced both basal and insulin-stimulated glucose uptake in MDA-MB-231 cells. Of note, the negative effect of hesperitin on basal glucose uptake was associated with GLUT1 downregulation, whereas the negative effect on insulin-induced glucose uptake was associated with impaired GLUT4 translocation to the cell membrane [37] (Table 3).

##### Quercetin and epigallocatechin-3-gallate (EGCG)

The flavonoids quercetin and EGCG (26 min) concentration-dependently inhibited glucose uptake by MCF-7 (IC_50_ = 11–23 µM) and MDA-MB-231 (IC_50_ = 44–16 µM) cells, respectively, associated with a decrease in lactate production. The effects of quercetin and EGCG were independent of estrogen signaling and did not involve Protein Kinase A (PKA), C (PKC), G (PKG) and calcium-calmodulin. A 4 h exposure to quercetin or EGCG induced also a decrease in glucose uptake, which was associated with an increase in GLUT1 transcription rates. Moreover, an antiproliferative and cytotoxic effect of both compounds was described in MCF-7 cells, which was more potent when extracellular glucose was present. So, inhibition of basal glucose uptake and consequently lactate production were concluded to be determinants of the cytotoxic and antiproliferative effects of quercetin and EGCG in breast cancer cells [40]. The inhibitory effect of quercetin on glucose uptake by breast cancer cells was confirmed in later studies, as shown next. Xintaropoulou et al. [57] verified that inhibition of growth of the HBL100 breast cancer cell line by quercetin (50–150 µM) is associated with an increase in the amount of extracellular glucose and a reduction in lactate production, suggesting inhibition of glucose uptake [57]. Quercetin was also found to decrease the mobility of MCF-7 and MDA-MB-231 cells, associated with a decrease glucose uptake, lactate production and GLUT1 protein levels [140], and to decrease glucose uptake and GLUT1 protein levels in MDA-MB-231 cells [141] (Table 3).

In relation to EGCG, a recent study using rodent 4T1 breast carcinoma cancer cells showed that EGCG inhibits breast cancer growth, both in vitro and in vivo, associated with a reduction in glucose and lactic acid levels and GLUT1 mRNA levels in these cells [142] (Table 3).

##### Phloretin and Phloridzin

The dihydrochalcone phloretin is found primarily in apples and pears and can also be produced when its glycoside phlorizin is consumed and subsequently nearly entirely converted into phloretin by hydrolytic enzymes in the small intestine. Several studies present evidence for an inhibitory effect of phloretin (and also its glycone phloridzin) in relation to glucose uptake by breast cancer cells. Phloretin and phloridzin were found to decrease glucose uptake by a rat breast adenocarcinoma cell line, both in vivo and in vitro [143]. Phloretin was also suggested to decrease glucose uptake (as assessed by the increase in the amount of extracellular glucose and the decrease in the amount of lactate produced) associated with an antiproliferative effect in HBL100, but not MCF-7 cell line [57]. Finally, inhibition of GLUT2 by phloretin was concluded to potentially suppress MDA-MB-231 cell growth and metastasis, although phloretin was found to increase GLUT2 protein levels. The authors concluded that phloretin treatment inhibited uptake of glucose, and, as a consequence, increased GLUT2 protein expression was required for cancer cell survival [63]. In relation to phloridzin, it reduced glucose uptake in several breast cancer cell lines, either alone [38,40] or associated with cytochalasin B [138] (Table 3).

##### Glabridin

This flavonoid decreased glucose uptake and lactate production, possibly mediated by a decrease in GLUT1 protein levels, in MDA-MB-231 cells [141] (Table 3).

##### (+)-Catechin

The anticancer efficacy of polyphenols can be enhanced by combining them with compounds such as amino acids and vitamins [144]. In this context, a catechin:Lys complex (Cat:Lys 1:2) was recently tested in MCF-7 and MDA-MB-231 breast cancer cell lines and in the non-tumorigenic breast (MCF12A) cell line. Cat:Lys (24 h) decreased glucose uptake and lactate production in MCF-7 cells but increased glucose uptake and lactate production in MDA-MB-231 and MCF12A cells. Cat:lys (24 h) was also found to increase GLUT1 mRNA expression levels in MDA-MB-231 cells. In contrast, a shorter-term exposure (26 min) of these cell lines to Cat:Lys caused an increase in glucose uptake in MDA-MB-231 and MCF12A cells but no effect on MCF-7 cells [145]. In contrast, (+)-catechin was found to increase glucose uptake by MCF-7 cells [138]. Moreover, in the work of Silva et al. [145], by using a GLUT inhibitor, it was concluded that: (a) there is a contribution of a GLUT-mediated mechanism in glucose uptake in the three breast cell lines, (b) Cat:Lys stimulates GLUT-mediated glucose uptake in MDA-MB-231 and MCF12A cell lines, and (c) Cat:Lys inhibits non-GLUT-mediated glucose uptake in MCF-7 cells, a conclusion that was supported by the results of GLUT1 mRNA expression. So, Cat:Lys shows no consistent effects on glucose uptake by the breast cell lines (Table 3). Thus, apparently, its antitumoral effect is not related to an effect on glucose uptake, because Cat:Lys showed a similar antiproliferative, cytotoxic, antimigratory, and proapoptotic effect on both cancer cell lines and a much less evident effect in the non-tumorigenic cell line [145].

##### Polyphenolic Esters

Zhang et al. [146] reported inhibition of basal glucose transport in MCF-7 cells and other cell lines (including H1299 lung cancer cell line) by synthesized polyphenolic esters. Although not tested in breast cancer cells, these basal glucose transport inhibitors also inhibited H1299 cell growth, and these two activities appear to be correlated (Table 3).

##### Curcumin

Very recently, the effect of curcumin (diferuloylmethane), a well-known phytopolyphenolic compound isolated from rhizome of the plant *Curcuma longa* was evaluated and this compound was found to reduce glucose uptake and lactate production in a variety of cancer cell lines, including in MCF-7 cells [147]. Curcumin was concluded to inhibit aerobic glycolysis by downregulating pyruvate kinase M2 expression, which drives the Warburg effect and thus is essential for survival of cancer cells. Nevertheless, a direct effect of curcumin on glucose transporters was not investigated [147] (Table 3).

##### Cardamonin

This chalcone, isolated from *Alpiniae katsumadai*, reduced glucose uptake as well as lactate production and efflux in the breast cancer MDA-MB-231 cell line [148] (Table 3).

##### Plant Extracts

Some studies have investigated the effect of plant extracts rather than the effect of individual compounds (Table 3). In one study, *Baeckea frutescens* leaves extracts were found to decrease glucose uptake in two breast cancer cell lines (MCF-7 and MDA-MB-231), associated with a cytotoxic and proapoptotic effect. Importantly, the extracts were devoid of cytotoxic effect in the non-tumoral breast cell line MCF10A and were also devoid of effect on glucose uptake [149]. In another study, an extract of *Petiveria alliacea* leaves and stems reduced glucose uptake and lactate production in the 4T1 breast cell line. However, glucose levels in supernatant, rather than direct measurement of glucose uptake, were measured [150]. Finally, an extract of Kudingcha leaves, one of the *Ligustrum robustum* species, was described to concentration-dependently reduce GLUT1 and GLUT3 protein expression levels and lactate production in two triple-negative breast cancer cell lines [151].

#### 5.3.2. Other Exogenous Natural Compounds

##### Cytochalasin B

The macrocyclic mycotoxin cytochalasin B is a known GLUT inhibitor used extensively in the literature of GLUT investigation. Cytochalasin is a GLUT1, GLUT2, and GLUT4 inhibitor [152,153]. This compound has been described to interfere also with glucose uptake in several breast cancer cell lines: T-47D [38,98] and MCF-7 and MDA-MB-231 [40] (Table 3).

##### Genipin

Genipin, an aglycone derived from an iridoid glycoside called geniposide extracted from gardenia fruits, caused a decrease in glucose uptake by the breast cancer cells T-47D and MDA-MB-435, although no effect on MCF-7 and MDA-MB-231 cells was found (Table 3). The effect of genipin was most pronounced in the T-47D cell line; in this cell line, an IC_50_ = 61 µM was calculated and a decrease in lactate production was also found. In this report, genipin was concluded to decrease cancer cell glucose uptake by reducing both glycolytic flux and mitochondrial oxidative phosphorylation, an effect that was related to inhibition of Uncoupling Protein 2 (UCP2)-mediated dissipation of energy and restriction of ROS production through proton leakage [154]. However, an effect of genipin on glucose transporters cannot be excluded.

##### Cantharidin

This sesquiterpenoid bioactive compound is secreted by beetles of the family of Meloidae [155]. Although it has been available for almost a century, its use has not been approved due to its high toxicity to the gastrointestinal tract. However, new anticarcinogenic properties have been found. In a recent study, cantharidin was found to inhibit aerobic glycolysis, associated with a decrease in GLUT1 protein expression levels. It was concluded that cantharidin inhibits nuclear translocation of pyruvate kinase isoform M2 (PKM2), which promotes the transcription of GLUT1. So, cantharidin interferes with the glycolytic metabolic loop between GLUT1 and PKM2 [155] (Table 3).

##### Betulinic Acid (BA)

BA is a natural pentacyclic terpene reported to be capable of inhibiting various malignancies. BA was recently reported to decrease the viability of breast cancer cell lines MCF-7 and MDA-MB-231, being ineffective against the non-malignant mammary epithelial cell line MCF-10A, indicating that BA might be a highly selective killing agent toward malignant cells. BA was shown to decrease glucose uptake and lactate production in the two cancer cell lines, but it was concluded that suppression of the glycolytic activity mainly occurred at the intracellular level, and no further investigation of its effect upon glucose uptake was done [156] (Table 3).

##### Benzyl Isothiocyanate (BITC)

Data from numerous preclinical studies advocate BITC, an aromatic isothiocyanate, which occurs naturally in edible cruciferous vegetables, promising for breast cancer chemoprevention [157]. This compound was described to increase glucose uptake by breast cancer cells, both in vivo and in vitro. This effect is probably a compensatory mechanism in response to inhibition of complex III of the mitochondrial respiratory chain and of oxidative phosphorylation caused by this compound [158]. Moreover the effect of BITC upon glucose uptake was found to be dependent on Akt activation (Table 3). So, these results indicate that BITC increased glucose uptake/metabolism in breast cancer cells and suggest that breast cancer chemoprevention by BITC may be augmented by pharmacological inhibition of Akt [159].

##### Docosahexaenoic Acid (DHA)

n-3 polyunsaturated fatty acids (PUFAs) have been proposed to have anticancer properties, and the effects on cancer cell metabolism constitutes one possible mechanism contributing to their anticancer effect [160]. The effect of DHA on breast cancer and non-cancer cell lines was evaluated (Table 3). It was concluded that DHA contributes to impaired cancer cell growth and survival by altering cancer cell metabolism, including by causing a decrease in glucose uptake, while not affecting non-transformed cells [161].

##### Vitamin D3 (VD3)

VD3, the bioactive form of Vitamin D, is known to be an important modulator of bone metabolism and diabetes, amongst many other effects. Low levels of VD3 are linked with an increased risk of cancer, whilst high levels of vitamin D3 usually promise better prognosis [162]. Although its positive effects have been shown, not much is known as to the mechanisms involved. Santos et al [163] tested the effect of VD3 on MCF-7 and MDA-MB-231 cells. VD3 significantly reduced GLUT1 mRNA and protein levels and glucose uptake in both cell types. Moreover, lactate production in the highly metastatic MDA-MB-231 cells was significantly reduced (Table 3). This study proved that VD3 decreased breast cancer cell viability along with reduced expression of GLUT1 and key glycolytic enzymes (hexokinase II and lactate dehydrogenase A), causing a decrease in glucose uptake.

##### γ-Tocotrienol

This member of the vitamin E family of compounds displays potent anticancer effects at doses having little or no effect on normal cell viability [164]. To test the role of glycolysis in the effect caused by γ-tocotrienol, Parajuli et al. [165] used the MCF-7 breast cancer cell line. γ-tocotrienol was found to decrease MCF-7 cell growth, coincident with a concentration-dependent decrease in glucose consumption and lactate production. This effect was correlated with a decrease in the expression of glycolytic enzymes (HK-II, PFK, PKM2, and LDHA) and MCT1, but the compound did not affect GLUT1 expression levels. So, GLUT1 reduction was not considered the mediator of the glycolysis inhibition caused by γ-tocotrienol [165] (Table 3).

##### d-Allose

This compound, a C-3 epimer of d-glucose with 80% of the sweetness of sucrose, exists in small quantities in nature. This compound is known to possess anticarcinogenic properties via upregulation of thioredoxin interacting protein (TXNIP) [166]. A study involving breast adenocarcinoma, hepatocellular carcinoma, and neuroblastoma cell lines concluded that d-allose downregulates GLUT1 expression and consequently glucose uptake, thus suppressing cancer cell growth, as a result of overexpression of TXNIP [166] (Table 3).

## 6. Effect of Stimulation of the Interaction of Anticancer Agents with GLUT

Conjugation of anticancer agents with glucose or other sugars is a widely exploited technique to design therapeutic agents, in order to improve their uptake into highly glycolytic cancer cells overexpressing GLUTs, thus increasing efficacy while reducing side effects. One possibility is to develop sugar-conjugated agents that can be transported into cancer cells through GLUT without inhibiting GLUTs themselves [168]. Another possibility is to promote interaction of anticancer agents with GLUT by their conjugation with an anti-GLUT antibody. Some of these agents have been tested in breast cancer cell lines, as shown next.

### 6.1. Adriamycin

Adriamycin (doxorubicin) is effective against many types of solid tumors in clinical applications. However, its use is limited because of systemic toxicity and multidrug resistance. Adriamycin conjugated with a glucose analogue (2-amino-2-deoxy-d-glucose) and succinic acid (2DG-SUC-ADM) was designed to target tumor cells through GLUT1, thus enhancing the selectivity of doxorubicin against cancer cells while reducing its toxicity to healthy cells [169]. In a work using several cancer cell lines, including MCF-7 and MDA-MB-231 cell lines, the complex showed better inhibition to tumor cells and lower toxicity to normal cells, and, most importantly, displayed a potential to reverse multidrug resistance. In vivo experiments also showed that this new complex could significantly decrease organ toxicity and enhance the antitumor efficacy compared with free ADM, indicating 2DG-SUC-ADM as a promising drug for targeted cancer therapy [169]. The GLUT1-mediated transport into the cells explained the specificity of 2DG-SUC-ADM, because uptake of free doxorubicin mainly occurred through diffusion, whereas the uptake of 2DG-SUC-ADM was mostly GLUT1-mediated [169].

Sztandera et al. [170] developed a glucose-modified PAMAM dendrimer for the delivery of doxorubicin (dox) to breast cancer cells, designed to specifically enter tumor cell with enhanced glucose uptake. They verified that PAMAM-dox-glucose conjugate exhibited pH-dependent drug release and an increased cytotoxic activity compared to free drug in MCF-7 cells, in the absence of glucose. They also verified that GLUT1 inhibition eliminated the toxic effect of the conjugate. So, they concluded that the cytotoxic effect of PAMAM-dox-glucose depends on presence of a functional GLUT1, suggesting specific, transporter-dependent internalization as a main route of cellular uptake of glucose-conjugated PAMAM dendrimers [170].

### 6.2. Paclitaxel

This drug is widely used for the treatment of breast, ovarian, and lung carcinomas, but its low water solubility severely reduces its clinical application. In this context, a new prodrug was designed to enhance its solubility and its selective delivery to cancer by a preferential uptake via GLUTs. More specifically, the glycoconjugation of paclitaxel led to a derivative in which the drug was linked to 1-methyl glucose via a short succinic acid linker. The resulting compound, whose transport was mediated at least in part by GLUT1, showed a comparable cytotoxicity against several cancer cells without toxicity on normal cells. Of note, paclitaxel linked to succinic acid resulted in a lower toxicity against MCF-7 cells than the parent compound, suggesting that the presence of glucose improved its cytotoxicity [171].

### 6.3. Oxiplatin

The platinum antitumor drug oxaliplatin is a commonly used chemotherapeutic agent; however, its multiple side effects severely limit its benefits. The conjugation with sugar portions was introduced as a strategy to improve the tumor-targeting ability of the drug and also to enhance its water solubility, allowing renal excretion and lower systemic toxicity. A glycosylated (trans-R,R-cyclohexane-1,2-diamine)-malonatoplatinum(II) derivative showed increased cytotoxicity compared to oxaliplatin in all the tested human carcinoma cell lines. Its potency was prevented when human colon cancer (HT29) and breast cancer (MCF-7) cells, which overexpress GLUTs, were treated with the GLUT inhibitor phlorizin, thus confirming that the uptake and the antiproliferative activity of this compound are GLUT-mediated [172].

In summary, a great potential of GLUT-mediated transport of therapeutics into cancer cells opens new roads for targeted delivery of anticancer drugs.

## 7. Conclusions and Future Perspectives

Despite the high-survival rate in breast cancer patients and the availability of well-designed and effective therapeutic strategies, especially for hormone receptor or HER2-positive breast cancer, more drug research is still needed, particularly regarding triple-negative breast cancer, because of its unresponsiveness to hormone or anti-HER2 therapy. In this context, metabolic targeting of tumors, and more specifically targeting GLUT1-mediated glucose transport, constitutes an interesting approach. In addition to GLUT1, there are a several other potential cancer therapies that target the cellular energetic metabolism pathway in tumors. Indeed, many compounds targeting energy metabolism are currently in trial or approved as therapeutic agents for cancer [48,49]. Preclinical data from these inhibitors are encouraging; therefore, they represent additional options for targeting the enhanced aerobic glycolysis in cancer.

In this review, we show that a wide range of compounds, ranging from endogenous to dietary compounds and synthetic compounds, are able to interfere with glucose uptake by breast cancer cells. Moreover, for some of the presented compounds, their antitumoral effect is concluded to result from the effect on glucose uptake. Since cancer cells are highly dependent on glucose, even if these compounds possess other anticancer-inducing mechanisms, their effect on glucose uptake certainly contributes to their antitumoral effect.

The mechanisms underlying the modulatory effect of the compounds upon glucose uptake are very diverse, ranging from a direct effect upon the transporter, inhibition of transporter gene expression or protein synthesis, impairment of membrane insertion of the transporter, and redox balance modulation. Additionally, the effect of compounds on glucose uptake may be secondary to a decrease in the activity of glycolytic enzymes or signaling pathways. However, it should be pointed out that, for most of the presented compounds, the mechanisms underlying their modulatory effect upon glucose uptake by breast cancer cells have not been investigated. So, more research is needed in this area. A better knowledge of the mechanisms able to interfere with glucose transporter function in cancer cells may open new windows for therapeutic targets in breast cancer.

Most studies on GLUTs in cancer are focused on their role and regulation in the tumor cells. However, solid tumors are composed of several cell types, forming a dynamic and complex network. In this context, a role for GLUT1 in glycolytic reprograming enabling survival, growth, and expansion of effector T lymphocytes has been demonstrated [173]. T lymphocytes in tumors constitute a primary cellular target for immunotherapies including adoptive T cell therapy and immune checkpoint blockade. Moreover, GLUT1 induction was observed in human fibroblasts placed in contact with prostate cancer cells [174]. Cancer-associated fibroblasts (CAFs) are known to promote tumor growth, invasion, chemoresistance, and angiogenesis. So, the requirements for glucose entry and usage by cancer-supporting or cancer-antagonizing cells add on to the complexity of metabolic rewiring in cancer.

Metabolic targeting of tumors using GLUT inhibitors has attracted more and more attention in the past years, which can be demonstrated by the growing number and more recent publications on this subject. Most therapeutic strategies that are being developed to target GLUTs in cancer are in the preclinical phase of drug development. These preclinical data suggest that inactivation of GLUT1, leading to glucose starvation that ultimately leads to cell death, is a viable drug target for cancer therapy [52,175].

Regrettably, therapies designed to target this pathway have not been fully translated to the clinic yet, and clinical trials in cancer patients using GLUT inhibitors to ensure their safety and/or efficacy are still largely lacking. One of the major obstacles to the success of GLUT1-based therapies is the potential systemic toxicity, because GLUT1 is ubiquitously expressed in mammalian tissues. Although it is expected that targeting GLUT1-mediated glucose uptake will have a much more marked effect on cancer cells than in non-cancer cells, because cancer cells are much more sensitive to glucose deprivation than normal cells, a certain degree of side effects may be expected, especially those occurring in organs characterized by high glucose-consumption rates such as the brain, immune system and stem cells. One example is the observation that several glucose transport inhibitors, tested in phase I clinical trials for hepatocellular and prostate cancer, were associated with significant side effects [176]. So, selective blockade of GLUTs in tumor cells still remains a key challenge and research on this subject should be fostered in the near future.

In this context, important points should be considered in future research:(1).Development of cancer-specific and potent GLUT inhibitors that minimize side effects;(2).Development of selective targeted delivery of GLUT inhibitors (using recent imaging technology) directly into the tumor or intra-arterially near the tumor, or micro-encapsulation of the inhibitor [52];(3).More studies on GLUT regulation and function in vivo should be conducted;(4).Studies on multi-targeted inhibition, allowing lower doses of GLUT inhibitors to be used. For instance, the preliminary results from clinical trials of 2-DG as a monotherapy are inconclusive and ambiguous, with toxic and side effects being reported [49,177]. Currently, 2-DG was reintroduced for use in combination approach, as reported in more recent preclinical and clinical studies, using 2-DG at lower doses to produce synergistic anticancer effects with other chemotherapeutic agents or irradiation. Thus, combination treatments using 2-DG may have encouraging outcomes providing a new opportunity for cancer combination therapy [52,175];(5).An accurate verification and analysis of the transporter expression profiles, because cancers are extremely heterogeneous diseases and they have unique metabolic features. Moreover, abnormal glycolysis due to defects of mitochondrial oxidative phosphorylation is not absolutely common in spontaneous tumors [8,178];(6).Many natural compounds interfere with glucose uptake at higher than dietary/physiological concentrations (e.g., polyphenols). For some dietary compounds, these concentrations are not attainable even after the consumption of food supplements due to reduced bioavailability and/or metabolism of these compounds. For these compounds (a) improvement of their bioavailability and delivery, as it would result in the improvement of their biological effects in vivo, (b) the use of natural or synthetic analogs that have better bioavailability or more potency, or (c) combination with other glucose transport inhibitors or with conventional therapy, resulting in a synergistic effect or in improvements of its bioavailability [144,179] are possible strategies.(7).Resistance resulting from the treatment with GLUT inhibitors, involving development of different routes for energy supply using other energy substrates, is another issue that should not be underestimated. Indeed, cancer cells display metabolic plasticity and can overcome the inhibition of a specific metabolic pathway via the expression or up-regulation of alternative pathways. Moreover, adjacent cells such as fibroblasts can offer metabolic intermediates for the needs of cancer cells. To overcome this problem, combination of other targeted therapy drugs with GLUT inhibitors may be an effective way [180]. Alternatively, combination of two or more antimetabolic agents inhibiting different metabolic pathways simultaneously would decrease resistance and prevent relapse [52].(8).Research focused on the design of GLUT-transportable chemotherapeutics, which may provide therapeutic selectivity [180];(9).Characterization of glucose transporters in tumors treated with immunotherapies, to determine if their expression in tumor cells or cells of the tumor environment changes upon treatment, and if their activity is linked to the outcome of therapy [181];(10).Research focused on overcoming side effects that are expected, especially those occurring in organs such as the brain. In this context, it is known that, in starvation, ketone bodies can replace glucose as fuel for the brain. Therefore, a combined administration of GLUT-interfering agents with either a ketogenic diet or dietary supplements such as triheptanoin (which is currently being tested for the treatment of GLUT1 deficiency [182]), should improve the safety profile of these compounds [53].

In conclusion, in this review we show that several chemically distinct compounds interfere with glucose uptake by breast cancer cells, and these GLUT inhibitors should be used as starting point in future research, which should focus in developing new compounds/combinations/delivery methods to solve specific problems already identified.

## Figures and Tables

**Table 1 cancers-12-00154-t001:** Effect of synthetic compounds on glucose uptake by breast cancer cells.

Compound	Concentration (Time)	Cell Line/Model	Effect	Mechanism of Action	Ref
GLUT inhibitors					
WZB117	3–30 µM (24 h)	MCF-7, HBL100	↑ extracellular glucose levels ↔ (HBL100) or ↓ (MCF-7) extracellular lactate levels	-	[57]
WZB117	0.6 µM (16 h)	MCF-7, MDA-MB-231	↓ glucose uptake, lactate production and extracellular levels	↓ GLUT1 mRNA and protein levels	[58]
STF31	0.01–1 µM (24 h)	MCF-7, HBL100	↓ glucose and ↔ lactate in the extracellular medium (HBL100) ↑ glucose and ↓ lactate in the extracellular medium (MCF-7)	-	[57]
WZB27 and WZB115	30 µM and 10 µM, respectively (15 min)	MCF-7	↓ glucose uptake	-	[56]
2-deoxy-D-glucose	2 mM (24 h)	MCF-7 MDA-MB-231	↔ glucose uptake ↓ glucose uptake	- -	[65]
GLUT4 shRNA	transfection	MCF-7	↓ glucose uptake	-	[41]
GLUT1 shRNA	transfection	SK-BR3, MDA-MB-468	↓ glucose uptake and lactate production	-	[66]
GLUT1 shRNA	transfection	78617GL	↓ glucose uptake, consumption and lactate production	-	[27]
GLUT1 shRNA	transfection	MDA-MB-231, Hs578T	↓ glucose uptake	-	[67]
BAY 876	3 µM (5 days)	MDA-MB-436	↓ glucose uptake	-	[63]
Anti-GLUT1 antibody	0.1 mg/mlL (18 h)	MDA-MB-231	↓ glucose uptake	-	[69]
Anti-diabetics					
Metformin	1–10 mM (24–48 h)	MDA-MB-231 Orthotopically implanted MDA-MB-231 cells in mice	↔ glucose uptake	↔ GLUT1, GLUT2, GLUT3, and GLUT4 mRNA levels	[73]
Metformin	0.05–5 mM (26 min) 0.5–1 mM (24 h)	MCF-7, MDA-MB-231	↓ glucose uptake ↑ glucose uptake and lactate production	- ↔ GLUT1 mRNA levels	[65]
Metformin	10 mM (12 h)	MCF-7	↔ GLUT1 and SGLT1 protein levels	↓ PPARδ agonist-induced ↑ glucose uptake, consumption, lactate production and GLUT1 and SGLT1 mRNA and protein levels	[75]
Metformin	2 mM (8 weeks)	MDA-MB-231, MDA-MB-436	↑ glucose uptake	-	[74]
Phenformin	2 mM (12 h)	MDA-MB-231	↑glucose uptake, lactate production	↑ GLUT1 mRNA and protein levels NBR2-dependent	[77]
Troglitazone	20 µM (1 h)	MCF-7, MDA-MB-231, MDA-MB-468, T47D	↑ glucose uptake and lactate production	↔ GLUT1 protein levels PPARγ-independent, MAPK-, AMPK-, and EGFR-dependent	[79]
Chemotherapeutic agents					
Cisplatin	20 µM (48 h)	MDA-MB-231	↓ glucose uptake, lactate production and GLUT1 and GLUT4 mRNA levels	↓ integrin β5/FAK signaling pathway ↓ GLUT1 and GLUT4 mRNA levels	[36]
Sorafenib	7.5 µM (6–24–48 h)	MCF-7 MDA-MB-231 SKBR3	6 h: ↑ glucose uptake, utilization, lactate production (MCF-7 and SKBR3), no effect (MDA-MB-231) 24 h: ↓ glucose uptake and utilization (MCF-7 and SKBR3), ↑ glucose uptake and utilization (MDA-MB-231)	6 h: ↑ GLUT1 protein levels (MCF-7 and SKBR3), no effect (MDA-MB-231) 24 h: ↓ GLUT1 protein levels (MCF-7 and SKBR3), ↑ GLUT1 protein levels (MDA-MB-231) 48: ↓ GLUT1 protein levels (all cell lines) AMPK-dependent inhibition of mTORC1 pathway	[82]
Trastuzumab	initial dose of 4 mg/kg and 2 mg/kg on the 8^th^ and 15^th^ day	xenografts derived from HER2-overexpressing MDA-MB-453 human breast tumour grown in SCID mice	↓ glucose uptake	↓ GLUT1 protein levels	[83]
Doxorrubicin	1 μM (24 h); analysis for 3 d	MCF-7	↓ glucose uptake	↑ GLUT1 mRNA and protein levels	[84]
5-Fluorouracil	200 µM (24 h); analysis for 3 d	MCF-7	↓ glucose uptake	↑ GLUT1 mRNA and protein levels	[84]
Selenium and doxorubicin, free and nanoparticles	Se, nano-Se: 10 µM (24 h) DOX, nano-DOX: 50 µM (24 h)	MCF-7, MDA-MB-231	↑ extracelular glucose levels	-	[85]
Palbociclib + BYL719	0.5 μM palbociclib and 5 μM BYL719 alone or in combination; normoxic or hypoxic conditions (24 h)	MDA-MB-231	Alone: ↓ glucose uptake and consumption When combined: enhancement of effect, in both normoxic and hypoxic conditions	Alone: ↓ GLUT1 protein levels When combined: enhancement of effect, in both normoxic and hypoxic conditions The greater efficacy of the combination ascribed to inhibition of both PI3K/mTOR signaling and c-myc expression	[88]
Palbociclib + paclitaxel	0.5 μM palbociclib and 10 nM paclitaxel alone (48 h) or palbociclib (24 h) + paclitaxel (24 h); normoxic or hypoxic conditions	MDA-MB-231	Alone: ↓ glucose uptake and consumption When combined: enhancement of effect, in both normoxic and hypoxic conditions	Alone: ↓ GLUT1 protein levels When combined: enhancement of effect, in both normoxic and hypoxic conditions The greater efficacy of the combination ascribed to enhancement of inhibition of Rb/E2F/c-myc signaling	[87]
Others					
Tamoxifen	2 µM (72 h)	MCF-7	-	↓ of the ↑ in GLUT1 protein levels induced by E2	[33]
Saracatinib	0.5–1 µM	MCF-10A, MCF12A, with or without HER-2 overexpression	↓ glucose uptake	↓ of ERK1/2-MNK1-eIF4E-mediated cap-dependent translation of c-Myc and transcription of the glucose transporter GLUT1	[93]
RITA, nutlin 3a (P53 activating compounds)	1 µM (8 h)	MCF-7	-	↓ GLUT1 mRNA levels Not related to induction of apoptosis	[94]
Pifithrin-α (PFTα; P53 inhibitor)	Not mentioned	MCF-7	-	↑ GLUT1 mRNA levels, abolishes the effect of RITA on GLUT1 mRNA levels Not related to apoptosis induction	[94]
Wortmannin (PI3K inhibitor)	100 nM (30 min)	T-47D	↔ glucose uptake	-	[98]
Akt1i, Akt2i, Akt1/2i (Akt inhibitors)	10 µM (30 min)	T-47D	↓ glucose uptake	Akt signaling pathway-independent	[98]
PGC1b or HKDC1 shRNA	transfection	MCF-7	↓ glucose uptake	-	[102]
miRNA-34a inhibitor	transfection	BT-20, MDA-MB-231	↓ glucose uptake and GLUT1 protein levels	Inhibition of miRNA34a	[103]
agomiR-186-3p	Systemic delivery of cholesterol-modified agomiR-186-3p	Mice bearing tamoxifen-resistant breast tumors	↓ tumor growth and ^18^F-FDG uptake	EREG (agonist of EGFR)-mediated	[104]

**Legend:** ↑, increase; ↓, decrease; ↔ no effect; - not studied.

**Table 2 cancers-12-00154-t002:** Effect of endogenous compounds on glucose uptake by breast cancer cells.

Compound	Concentration (Time)	Cell Line/Model	Effect	Mechanism of Action	Ref
Hormones					
Melatonin	1 mM (24 h)	MCF-7, MDA-MB-231	-	↓ GLUT1 protein levels (normal and acidic conditions)	[108]
17β-estradiol	3 × 10^−8^ M (7 days)	T47D-clone 11	↑ glucose utilization	-	[110]
17β-estradiol	10 nM (24 h)	ZR-75-1	↑ glucose (0.1 mM) uptake ↔ glucose (15 mM) uptake	↔ GLUT1, GLUT2, GLUT3 and GLUT4 mRNA levels ↔ GLUT1, GLUT2, GLUT3 protein levels, ↑ GLUT4 protein levels	[30]
17β-estradiol	3 × 10^−8^M (72 h)	MCF-7, T47D, ZR-75-1	-	↑ GLUT1 protein levels	[33]
17β-estradiol	10 nM (24 h)	T-47D MDA-MB-231 and MDA-MB-468	↑ glucose uptake and lactate production ↔ glucose uptake	↔ GLUT1 protein levels ER- and PI3K–Akt-dependent Non-genomic, membrane-initiated action	[111]
17β-estradiol	10 nM (25–45 min)	MCF-7	↑ glucose uptake	↑ GLUT4 in plasma membrane (but not total protein levels) ↔ GLUT1 (total and plasma membrane protein levels) ERα- and PI3K-dependent Non-genomic, membrane-initiated action	[35]
17β-estradiol	100 nM (48 h)	MCF-7	↔ glucose uptake	↔ GLUT1 mRNA levels	[109]
Progesterone	10 nM (24 h)	ZR-75-1	↑ glucose (0.1 mM) uptake ↔ glucose (15 mM) uptake	↑ GLUT1 and GLUT3 mRNA levels, ↔ GLUT2 and GLUT4 mRNA levels ↑ GLUT1, GLUT3, and GLUT4 protein levels, ↔ GLUT2 protein levels	[30]
Dexamethasone	1 µg/mL (2 weeks)	MCF-7, MDA-MB-231	Small ↑ in glucose uptake	↑ GLUT4 mRNA levels	[112]
KL-1	Not mentioned	MCF-7	↓ glucose uptake and lactate production	-	[117]
Insulin	40 µg/mL (8 h)	MCF-7	-	↑ GLUT1 and GLUT3 protein levels	[120]
Others					
Lactic acid	10–25 mM (48 h)	Single cells isolated from 4T1 and MDA-MB-231 tumors grafts in mice 4T1 and MDA-MB-231	Single cells from the core of tumors grafts took up less glucose than those from the periphery Lactic acid levels were higher in the core of the tumor ↓ glucose uptake	-	[123]
Lactic acid	10 mM (26 min)	T-47D	↑ glucose uptake	-	[38]
Epidermal growth factor	100 ng/mL (24 h)	T-47D, MDA-MB-468, MCF-7	↑ glucose uptake (T-47D, MDA-MB-468), ↔ glucose uptake (MCF-7) ↑ lactate production (T47D)	↑ GLUT1 protein levels (T47D) PI3 kinase (PI3K) activation	[127]

**Legend:** ↑, increase; ↓, decrease; ↔ no effect; - not studied.

**Table 3 cancers-12-00154-t003:** Effect of exogenous compounds on glucose uptake by breast cancer cells.

Compound	Concentration (Time)	Cell Line/Model	Effect	Mechanism of Action	Ref
Polyphenols					
Gossypol	10 µM (25 h)	MCF-7	↑ glucose consumption and lactate production ↔ ratio lactate produced/glucose consumed	Quasi-competitive inhibition	[135]
Naringenin	10 µM (15 min)	MCF-7, T-47D	↓ basal and insulin-stimulated glucose uptake	Inhibition of PI3K, Akt, and MAPK pathways	[39]
Genistein	10–100 µM (10 min)	MCF-7, MDA-MB-231	↓ glucose uptake	-	[136]
Genistein, daidzein and a soy seed extract	22 µM, 52 µM and 166 µg/mL, respectively (72 h)	MCF-7, MDA-MB-231	↓ glucose uptake	-	[137]
Myricetin, resveratrol genistein, kaempferol	10–100 µM (26 min)	MCF-7	↓ glucose uptake	Mixed-type inhibition for kaempferol	[138]
Catechin	100 µM (26 min)	MCF-7	↑ glucose uptake	-	[138]
Resveratrol	150 µM (24 h)	T-47D	↓ glucose uptake	↓ GLUT 1 protein levels ↓ intracellular ROS causing ↓ of HIF-1α accumulation	[139]
Hesperetin	50–100 µM (24 h)	MDA-MB-231	↓ basal glucose uptake ↓ insulin-stimulated glucose uptake	↓ GLUT 1 mRNA and protein levels ↓ GLUT4 cell membrane translocation	[37]
Quercetin and EGCG	1–500 µM (26 min) 1–100 µM (4 h)	MCF-7, MDA-MB-231	↓ glucose uptake	Competitive, independent of PKA, PKC, PKG, and calcium-calmodulin intracellular pathways	[40]
Quercetin, Phloretin	50–150 µM (24 h)	HBL100	↓ glucose uptake	-	[57]
Quercetin	30 µM (6.5 h)	MCF-7, MDA-MB-231	↓ glucose uptake, ↓ lactate production	↓ GLUT1 protein levels	[140]
Quercetin	Not mentioned	MDA-MB-231	↓ glucose uptake, ↔ lactate production	↓ GLUT1 protein levels	[141]
Glabridin	Not mentioned	MDA-MB-231	↓ glucose uptake, ↓ lactate production	↓ GLUT1 protein levels	[141]
Phloretin and phloridzin	Not mentioned	Rat mammary adenocarcinoma	↓ glucose uptake in vitro and in vivo	-	[143]
Phloretin	10–150 µM (16 h)	MDA-MB-231	-	↑ GLUT2 protein levels	[167]
Phloridzin	1 mM (26 min)	T-47D	↓ glucose uptake	-	[38]
Phloridzin	0.5–1 mM (26 min)	MCF-7, MDA-MB-231	↓ glucose uptake	-	[40]
Phloridzin + Cytochalasin B	0.5 mM and 50 µM, respectively (26 min)	MCF-7	↓ glucose uptake	-	[138]
EGCG	20–240 µM (24 h) 20–80 µM (24 h) 20 mg/kg (mice)	4T1	↓ glucose and lactic acid levels ↓ glucose and lactic acid levels in tumour	↓ GLUT1 mRNA levels	[142]
Cat:Lys (1:2) complex	0.01–5 mM (26 min)	MCF-7, MDA-MB-231 and MCF12A	↑ glucose uptake (MDA-MB-231 and MCF12A) ↔ glucose uptake (MCF-7)	-	[145]
Cat:Lys (1:2) complex	0.01–1 mM (24 h)	MCF-7, MDA-MB-231 and MCF12A	↑ glucose uptake and lactate production (MDA-MB-231 and MCF12A) ↓ glucose uptake and lactate production (MCF-7)	↑ GLUT1 mRNA levels (MDA-MB-231) ↔ GLUT1 mRNA levels (MCF-7 and MCF12A)	[145]
Compound 1a and 2a (polyphenolic esters)	30 µM (15 min)	MCF-7	↓ glucose uptake	-	[146]
Curcumin	20 µM (24 h)	MCF-7	↓ glucose uptake and lactate production	-	[147]
Cardamonin	20–80 µM (6 h)	MDA-MB-231	↓ glucose uptake and lactate production	-	[148]
*Baeckea frutescens* extracts	IC50 = 10–127 μg/mL (MCF-7 cells; 72 h)	MCF-7, MDA-MB-231	↓ glucose uptake	-	[149]
*Petiveria alliacea* extract	3 µg/mL (48 h)	4T1	↓ glucose uptake	-	[150]
Kudingcha extract	100–200 µg/mL (48 h)	MDA-MB-231, HCC1806	- Decrease in lactate levels	↓ GLUT1 and GLUT3 protein levels	[151]
Others					
Cytochalasin B	25 µM (30 min)	T-47D	↓ glucose uptake	-	[98]
Cytochalasin B	50–100 µM (26 min)	T-47D	↓ glucose uptake	-	[38]
Cytochalasin B	10–50 µM (26 min)	MCF-7, MDA-MB-231	↓ glucose uptake	-	[40]
Cantharidin	0.5–2 µM (24 h)	MCF-7, MDA-MB-231	↓ glucose consumption and lactate production	↓ GLUT1 protein levels	[155]
Betulinic acid	5–40 µM (3 h)	MCF-7, MDA-MB-231	↓ glucose uptake and lactate production	-	[156]
Genipin	50–100 µM (24 h)	T-47D, MDA-MB-435 MCF-7, MDA-MB-231	↓ glucose uptake ↔ glucose uptake	-	[154]
Benzyl isothiocyanate	3 mmol/kg diet 2.5 µM (24 h)	MMTV-neu mice mammary tumours MDA-MB-231, SUM159	↑ glucose uptake ↑ glucose uptake	↑ GLUT1 total and membrane-located protein levels Dependent on Akt activation	[159]
Docosahexaenoic acid	100 µM (48 h)	MDA-MB-231, BT-474, MCF-10A	↓ glucose uptake and metabolism (MDA-MB-231, BT-474) ↔ glucose uptake and metabolism (MCF10A)	-	[161]
Vitamin D_3_	0.5–1 µM (24 h)	MCF-7, MDA-MB-231	↓ glucose uptake (both cell lines) ↓ lactate production (MDA-MB-231 cells)	↓ GLUT1 mRNA and protein levels (both cell lines) AMPK-dependent inhibition of mTOR (?)	[163]
γ-Tocotrienol	6–8 µM (96 h)	MCF-7	↓ glucose consumption and lactate production	↔ GLUT1 mRNA levels	[165]
d-allose	50 mM (7 days)	MDA-MB-231	-	↑ GLUT1 mRNA and protein levels Mediated by overexpression of TXNIP	[166]

**Legend:** ↑, increase; ↓, decrease; ↔ no effect; - not studied.

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
