# Peer review of "Targeting Glucose Transporters for Breast Cancer Therapy: The Effect of Natural and Synthetic Compounds"

_cancers, 2020, doi:10.3390/cancers12010154_

Round 1
Reviewer 1 Report
In this review, the Authors aimed to discuss the natural and synthetic compounds that interfere with the uptake of glucose by breast cancer cells, thus may act as anticancer agents due to the ability to inhibit tumor progression.
The topic is interesting and the information timely. As stated by the Authors GLUT expression correlates with a high malignant potential, invasiveness and consequently a poor prognosis. Therefore, it could be interesting if the Authors would provide mechanistic data regarding the regulation of GLUT genes in breast malignancies.
Author Response
Thank you very much for the helpful comments. Shown next, we changed the manuscript according to your criticisms, and we think it has improved.
Reviewer 1
It could be interesting if the Authors would provide mechanistic data regarding the regulation of GLUT genes in breast malignancies.
In agreement with your proposal, this information was added (lines 102-111). The text concerning GLUT1 upregulation in cancer was also slightly changed, for a better coherence (lines 87-101).
Reviewer 2 Report
This manuscript is a review of what is known regarding glucose transport in response to breast cancer therapy. The authors discuss the general phenomenon of increased glucose transport in cancer and the rationale behind these transporters being therapeutic targets. The bulk of the review focuses on drugs that directly target glucose transport, either synthetic exogenous compounds, natural exogenous compounds, and natural endogenous effectors of glucose transport. The last section focuses on the effect of targeting cancer cells by targeting glucose transporters. The following comments are intended to improve the manuscript:
Comments:
There are several sections of the manuscript where there are very short paragraphs (≤3 sentences) but the adjacent paragraphs are part of the same point the authors are trying to make. It may be easier for readers if the manuscript was edited and some of these paragraphs joined together. The tables includes are helpful in their level of detail but are very cumbersome to consume. Perhaps some thought should be given to restructuring them to make it easier to take in. While the current version of the review is substantial in length, something helpful to add may be a better description in the introduction of how glucose transporters function normally.Author Response
Thank you very much for the helpful comments. Shown next, we changed the manuscript according to your criticisms, and we think it has improved.
Reviewer 2
There are several sections of the manuscript where there are very short paragraphs (≤3 sentences) but the adjacent paragraphs are part of the same point the authors are trying to make. It may be easier for readers if the manuscript was edited and some of these paragraphs joined together.
In agreement with this criticism, various paragraphs were joined together, throughout the manuscript.
The tables includes are helpful in their level of detail but are very cumbersome to consume. Perhaps some thought should be given to restructuring them to make it easier to take in.
Following your comment, we made some changes in the tables that we think make them easier to read.
While the current version of the review is substantial in length, something helpful to add may be a better description in the introduction of how glucose transporters function normally.
In agreement with your criticism, this information was added (lines 69-74 and 79-83).